# Processive chitinase is Brownian monorail operated by fast catalysis after peeling rail from crystalline chitin

Akihiko Nakamura [1,2], Kei-ichi Okazaki [1], Tadaomi Furuta [3], Minoru Sakurai[3] & Ryota Iino [1,2]

Processive chitinase is a linear molecular motor which moves on the surface of crystalline chitin driven by processive hydrolysis of single chitin chain. Here, we analyse the mechanism underlying unidirectional movement of *Serratia marcescens* chitinase A (SmChiA) using high-precision single-molecule imaging, X-ray crystallography, and all-atom molecular dynamics simulation. SmChiA shows fast unidirectional movement of ~50 nm s$^{-1}$ with 1 nm forward and backward steps, consistent with the length of reaction product chitobiose. Analysis of the kinetic isotope effect reveals fast substrate-assisted catalysis with time constant of ~3 ms. Decrystallization of the single chitin chain from crystal surface is the rate-limiting step of movement with time constant of ~17 ms, achieved by binding free energy at the product-binding site of SmChiA. Our results demonstrate that SmChiA operates as a burnt-bridge Brownian ratchet wherein the Brownian motion along the single chitin chain is rectified forward by substrate-assisted catalysis.

[1] Institute for Molecular Science, National Institutes of Natural Sciences, 444-8787 Okazaki, Aichi, Japan. [2] Department of Functional Molecular Science, SOKENDAI (The Graduate University for Advanced Studies), Hayama 240-0193, Japan. [3] Center for Biological Resources and Informatics, Tokyo Institute of Technology, Yokohama 226-8501, Japan. Correspondence and requests for materials should be addressed to A.N. (email: aki-naka@ims.ac.jp) or to R.I. (email: iino@ims.ac.jp)

Molecular motors convert various types of energies into unidirectional mechanical movement[1–3]. Most of the biomolecular motors working in the cell use adenosine triphosphate (ATP) as the chemical energy source. Recently, however, processive cellulase and chitinase have been rediscovered as molecular motors working in extracellular environments without using ATP[4,5]. Similar to a monorail car, they have long tunnel- or cleft-like polysaccharide-binding sites (Fig. 1a) and processively hydrolyse crystalline cellulose and chitin, major types of biomass on earth, to water-soluble disaccharides[6–9]. As a tool of biomass conversion to useful chemicals, they have been extensively studied[10–12].

Processive catalysis of cellulase and chitinase has an important role in efficient degradation of crystalline cellulose and chitin, respectively[13,14]. Structural studies have revealed significance of the line of aromatic amino acid residues at the polysaccharide-binding sites for processivity[15,16]. Although quantitative estimation of the processivity is a difficult task due to heterogeneous reaction at the liquid–solid interface, important insights have been obtained from unique biochemical assays[17]. Using fluorescent-labelled substrates[18,19] or an amperometric biosensor[20], the fraction of productively bound enzyme has been successfully determined to estimate the processivity. Recent applications of single-molecule imaging with high-speed atomic force microscopy[4,5,21] or total internal reflection fluorescence microscopy[22,23] have enabled to observe the productively bound moving enzymes directly.

In addition to the processive movements, single-molecule imaging has revealed that chitinase moves and, accordingly, produces disaccharides ten times faster than cellulase does[4,5], although the physical and chemical stabilities of crystalline cellulose and chitin are similar[24,25]. Therefore, the mechanism underlying fast unidirectional movement of chitinase is gaining attention. During the processive movement on crystalline chitin surface, chitinase keeps binding with single chitin chain in the catalytic cleft and repeats chemical and mechanical steps (Fig. 1b). In the chemical step, the catalytic amino acid residue and chitin $N$-acetyl group first cleave the glycosidic bond without attacking water, a mechanism called as substrate-assisted catalysis (Supplementary Figure 1)[26]. Then, the water molecule approaches the catalytic site, and the oxazolinium ion or oxazoline intermediate is hydrolysed. Chitobiose unit is removed from the chain end following the hydrolysis of the intermediate and the next chitobiose unit is peeled from the crystal surface (decrystallisation) accompanied with the forward step. Considering the size of the reaction product, chitobiose (1.04 nm, disaccharide length), chitinase is expected to move with 1 nm step sizes.

To understand operation mechanisms driving the unidirectional motion of processive chitinase, we analyse elementary steps of movement coupled with processive catalysis, using high-precision single-molecule imaging probed with gold nanoparticle (AuNP). We verify fast unidirectional movement (~50 nm s$^{-1}$) with 1 nm forward and backward steps, consistent with the length of the reaction product, chitobiose. Analysis of the kinetic isotope effect (KIE) reveals that substrate-assisted catalysis is much faster than decrystallisation. Decrystallisation of single chitin chain is the rate-limiting step of movement, achieved by binding free energy at the product-binding site, and verified by X-ray crystallographic structural analysis and all-atom molecular dynamics (MD) simulations of the intermediate structures during sliding movement. Much larger forward-step ratio than backward step ratio is explained by the competition between the substrate-assisted catalysis and backward movement, indicating that the Brownian motion is rectified forward by fast catalysis. We demonstrate how chitinase controls the Brownian motion and extracts fast unidirectional movement for continuous degradation

of crystalline chitin without dissociation. The strategy evolved by chitinase can be applied to design fast-moving artificial molecular motors such as DNA walkers[3].

## Results

**Single-molecule imaging of chitinase movement with AuNP**. To achieve localisation precision and temporal resolution required for the detection of 1 nm steps of processive chitinase, we labelled the most studied chitinase A from a bacterium *Serratia marcescens* (SmChiA) with a 40 nm AuNP and observed scattering images using total internal reflection dark-field microscopy (TIRDFM; Fig. 1b)[27–29]. With our imaging system, the localisation precision of AuNP immobilised on glass surface was 0.3 nm at 0.5 ms temporal resolution (Supplementary Figure 2a, b). Distribution of signal intensity of AuNP immobilised on glass surface was well fitted by the single Gaussian function with peak ± SD of 0.29 ± 0.19 arbitrary unit (A.U., $N = 267$, Supplementary Figure 2c), indicating no aggregations of AuNP. Then, we analysed signal intensity, velocity, and run length of AuNP-labelled SmChiA molecules moving on the crystalline chitin. As crystalline chitin is also visible with our imaging system, but the signal intensity of AuNP was much higher than that of the crystalline chitin, binding and movement of AuNP-labelled SmChiA on the crystalline chitin were easily recognised (an example is shown in Supplementary Figure 3a). Furthermore, signal intensity of AuNP-labelled SmChiA (0.36 ± 0.10 A.U., mean ± SD, $N = 17$) was similar to that of the single AuNP immobilised on glass (Supplementary Figure 2c), indicating that each SmChiA molecule was labelled with single AuNP, not an aggregate. Small size of AuNP minimises the viscous drag of water affording observation of free and rapid motions. Similar velocities of AuNP- and Cy3-labelled SmChiA (51.9 ± 21.7 nm s$^{-1}$ ($N = 58$) and 55.3 ± 18.4 nm s$^{-1}$ ($N = 57$), mean ± SD respectively, Supplementary Figure 3b) indicated no adverse effects of AuNP labelling on movement. The average run length of SmChiA (86.2 nm, $N = 30$, Supplementary Figure 3c) was much shorter than the average length of crystalline chitin (3.5 μm, $N = 131$, Supplementary Figure 3d), suggesting the processivity of 83 (86.2 nm/1.04 nm) using the size of reaction product chitobiose (1.04 nm). This value is 2.3 times higher than the value (36 ± 5) estimated by biochemical assay[30], whereas comparable to that (60) estimated by previous single-molecule fluorescence imaging[23].

**Single-molecule imaging verified 1 nm stepping movement**. Figure 2a and b show examples of AuNP-labelled SmChiA movement and centroid trajectories. SmChiA molecules exhibited unidirectional movements along chitin fibrils (Supplementary Movie 1), showing transient pauses and abrupt movements (steps) in forward and backward directions (Fig. 2b and c, on-axis, green lines). A step-finding algorithm, applied to ± 2 ms median-filtered traces, objectively detected motion pauses and steps (Fig. 2b and c, step-fit, magenta lines, Supplementary Figure 4)[31]. Forward- and backward step size distributions were fitted by Gaussian super-positioning (Fig. 2d). Minimum forward (1.1 nm) and backward (−1.1 nm) step size peaks were detected, with a small fraction of double-sized (2.2 nm) forward steps. Ratios calculated from Gaussian areas were 69.3% (forward), 14.2% (double-sized forward), and 16.5% (backward). Figure 2e and Supplementary Figure 4e show on- and off-axis localisation precision distributions during detected pauses ($N = 997$) with 0.34 and 0.32 nm peak values, respectively. To verify experimental capability of detecting 1 nm steps, SmChiA trajectories were simulated and analysed using experimentally obtained localisation precisions, step sizes and ratio, and kinetic parameters (Supplementary Figure 5). Simulated trajectories resembled

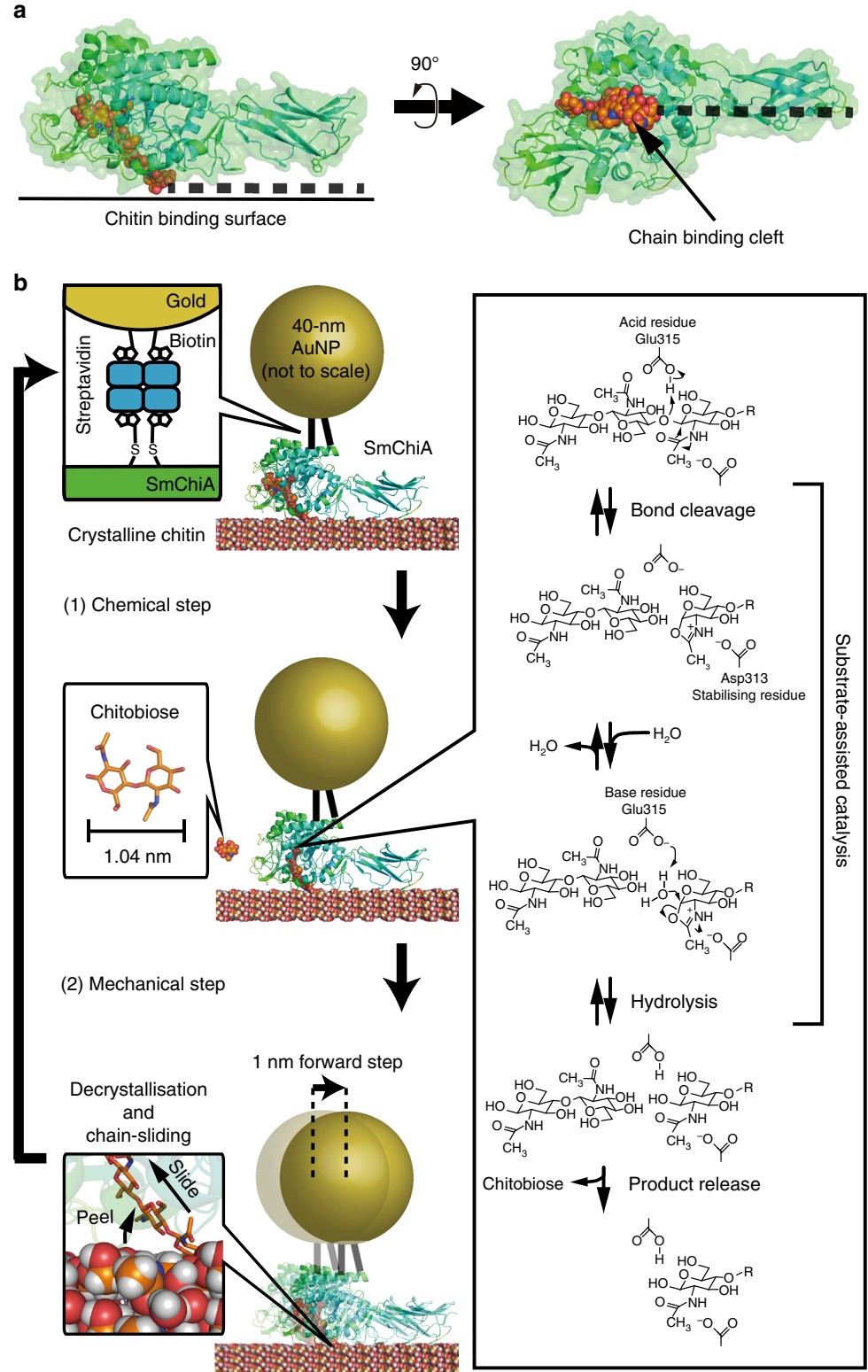

**Fig. 1** Schematic of single-molecule imaging of SmChiA probed by gold nanoparticle. **a** SmChiA has a flat surface for binding to crystalline chitin and the cleft for threading decrystallized single chain of chitin. **b** A single AuNP was attached to SmChiA via streptavidin-biotin interaction. Elementary steps were categorised as chemical (substrate-assisted catalysis and product release) or mechanical (decrystallization and chain sliding) step. Length of the reaction product chitobiose (1.04 nm) corresponds to expected step size of SmChiA movement

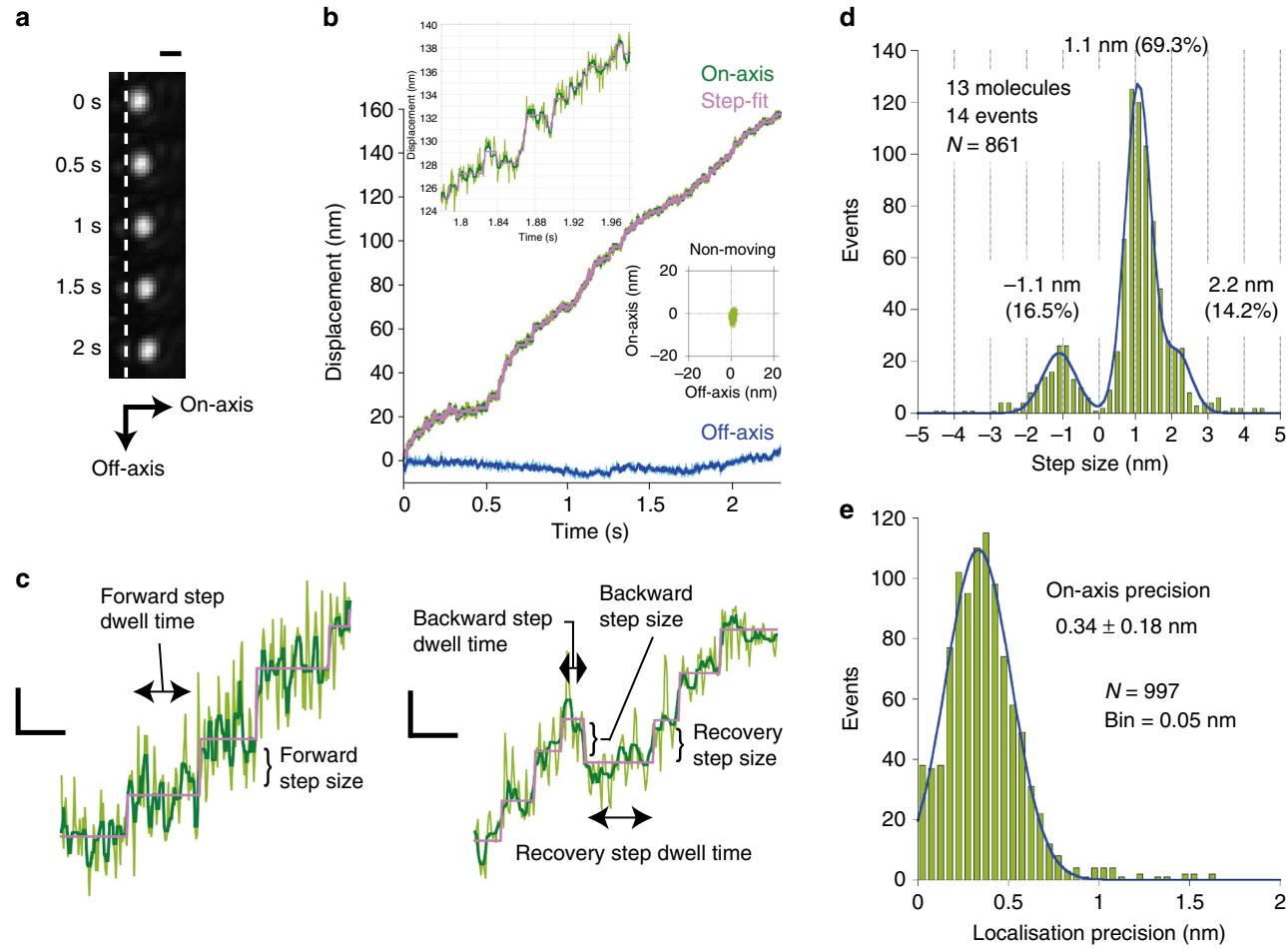

**Fig. 2** Stepping movement of SmChiA probed by 40-nm AuNP. **a** Example of image sequence. Scale bar is 400 nm. **b** Trajectory of SmChiA movement shown in **a**. Raw trajectory: light green; median-filtered (current ± 2 ms): dark green. Magenta line: steps and pauses fitted to median-filtered trajectory. Two-dimensional trajectory of non-moving molecule in the same field of view is also shown, in the same scale. **c** Examples of expanded trace. Forward, backward, and recovery step sizes and dwell times before these steps are indicated. Scale bars on vertical and horizonal axes are 1 nm and 30 ms, respectively. **d** Forward and backward-step size distribution. The peak values of Gaussian fittings (blue line) are shown. Step ratios were estimated from Gaussian areas. **e** On-axis localisation precision during pauses. The value is peak ± SD of Gaussian fitting (blue line)

experimental trajectories and step-finding algorithm correctly recovered the simulated steps before noise addition. Simulated time constants were well reproduced by the curve fitting of the dwell time distributions without the first bin (Supplementary Figure 6). Thus, experimental data were also fitted without the first bin (Supplementary Figure 7).

**Kinetic isotope effect revealed fast catalysis**. Next, we analysed distributions of pause length of moving SmChiA in detail. Dwell time distribution of the pauses before forward steps between 0 and 2 nm (Fig. 2c and d) was well fitted by a model assuming consecutive reaction with two time constants of $23.9 \pm 2.1$ and $2.9 \pm 1.0$ ms (Fig. 3a, parameters ± fitting errors). To identify the time constant corresponding to the substrate-assisted catalysis (glycosidic bond cleavage by the chitin $N$-acetyl group and the catalytic amino acid residue, followed by hydrolysis of oxazolinium ion (oxazoline) reaction intermediate), SmChiA movement was observed in $D_2O$, affording slower hydrolysis via the KIE (Fig. 3b, Supplementary Figure 8). This lengthened only the shorter time constant (3.5-fold; $10.1 \pm 2.3$ vs. $2.9 \pm 1.0$ ms, parameters ± fitting errors), indicating 2.9 ms in $H_2O$ as corresponding to the substrate-assisted catalysis, followed by product release, decrystallisation, and chain sliding requiring 23.9 ms.

Furthermore, similar magnitude of the KIE (2.3-fold) was observed with hydrolysis of a water-soluble substrate chitotetraose by a SmChiA mutant (W167A), which suppresses nonproductive binding of the substrate (Supplementary Figure 8e). This result verifies the notion that the KIE mainly occurs in the substrate-assisted catalysis, not in the decrystallisation process. Dwell time distributions before backward and subsequent recovery steps (Fig. 2c) were both fitted by single exponential decay with time constants of $18.3 \pm 0.6$ and $17.1 \pm 1.8$ ms, respectively (Fig. 3c and d, parameters ± fitting errors). Recovery step should comprise similar reactions as forward steps without product release. Accordingly, shorter time constant of recovery step (17.1 ms) than forward step (23.9 ms) was observed, indicating 6.8 ms (forward minus recovery) is the time constant of product release. Dwell time distribution before 2 nm forward steps (>2 nm in the step size distribution, Fig. 2d) was fitted by single exponential decay with time constant of $26.0 \pm 2.5$ ms (Fig. 3e, parameter ± fitting error). Similar time constant as 1 nm forward steps may arise, because only one chitobiose unit must be decrystallised owing to incomplete chitin surface crystallinity. Figure 3f shows summary of time constant and chemomechanical coupling scheme. Our results indicated that the rate-limiting step of SmChiA movement is not substrate-assisted

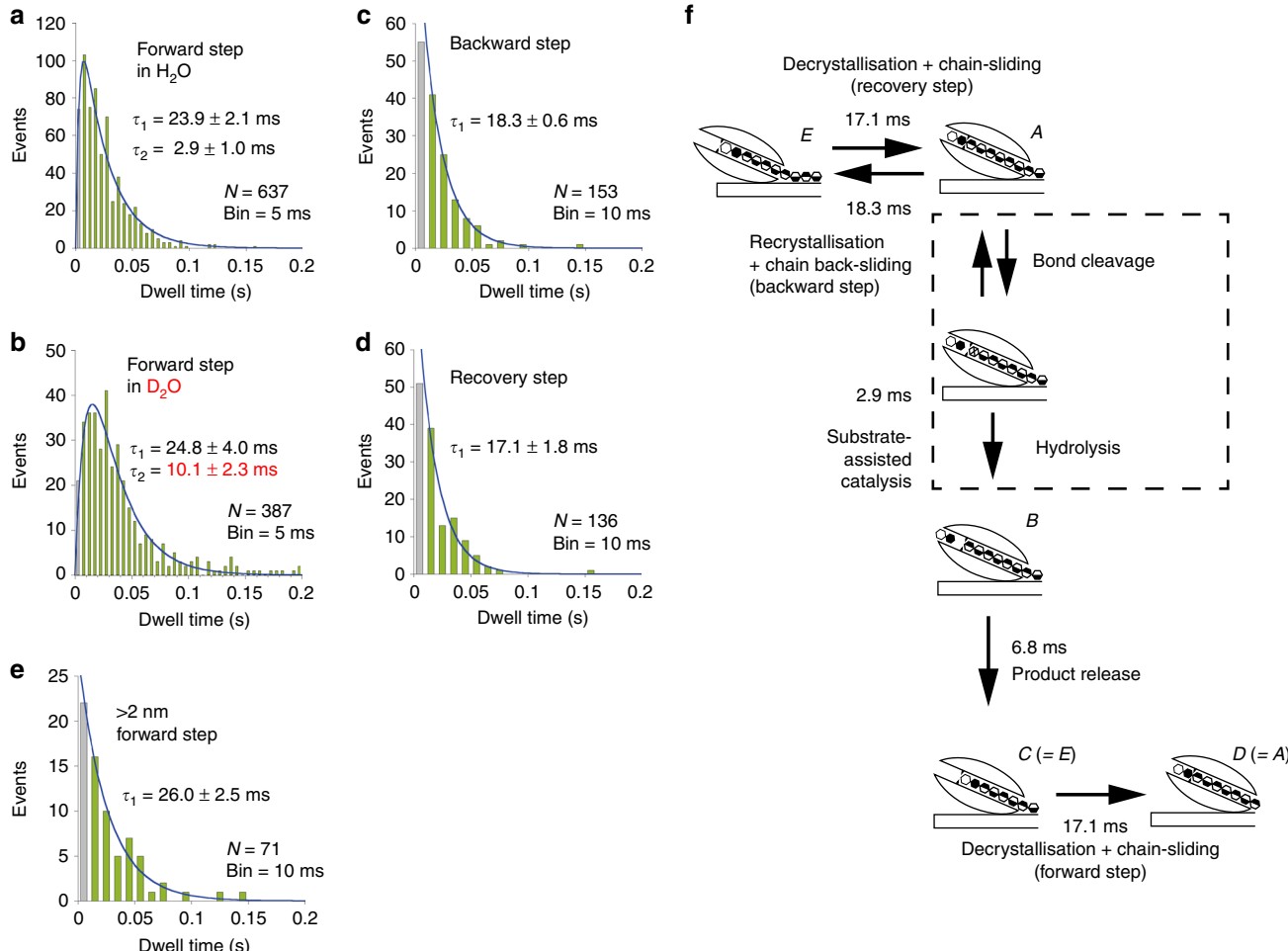

**Fig. 3** Dwell time analysis of SmChiA movement. **a**, **b** Dwell time distributions before 1 nm forward steps in $H_2O$ (**a**) and $D_2O$ (**b**) fitted by a model assuming consecutive reaction with two time constants (blue lines). **c–e** Dwell time distributions before backward (**c**), recovery (**d**), and before > 2 nm forward (**e**) steps fitted by single exponential decays (blue lines). The values are obtained parameters ± fitting errors. **f** Time constant summary of processive reaction cycle. Equilibrium constant between forward (state A) and backward (state E) stepped states is 1.1. Probabilities of substrate-assisted catalysis (86.3%) and backward step (13.7%) from state A are consistent with probabilities estimated by event number in Fig. 2d

catalysis and product release, but decrystallisation or chain sliding.

**Decrystallisation is the rate-limiting step of movement**. To investigate the mode of substrate binding and chain sliding, we generated an inactive mutant of SmChiA (D313A, K369M, F396A, W539A, and E540M). In this mutant, the amino acid residues constructing subsite +2 to −2 and stabilising the Michaelis complex were mutated. This mutant was designed to observe less stable substrate–enzyme complexes than the Michaelis complex. We solved four different crystal structures of the mutant bound with various lengths of oligosaccharides (Supplementary Figure 9 and Table 1). These crystal structures showed that the mutant can bind to the substrate at various positions. The mutated amino acid residues were not involved in the interactions between the substrate and the enzyme in the observed intermediate complexes (Supplementary Figure 10a). Furthermore, when the structures of intermediate substrates were superimposed with that of the wild-type SmChiA (PDB ID: 1EDQ), they did not show steric hindrances (Supplementary Figure 10b). Therefore, we concluded that the crystal structures unravelled in this study (Supplementary Figure 9) represent intermediate complexes during chain sliding.

Among these unravelled structures, one bound with chitohexaose (Fig. 4a) appeared to represent the intermediate state just before Michaelis complex formation, because the chitobiose unit at the chitohexaose reducing end was retained on Trp275, the same plane as the product-binding site of Michaelis complex-like structure of another inactive mutant (D313A, PDB ID: 1EIB, Fig. 4b). The Trp275 should be important to relieve steric hindrance at the product-binding site during sliding motion. The importance of Trp275 in processivity of SmChiA has been also reported previously[30], supporting this notion. Next, using this Sliding-intermediate complex as an initial structure, spontaneous sliding of chitin chain was analysed in all-atom MD simulations of the wild-type enzyme. We found that Glu315 showed double conformers in the crystal structure of the Sliding-intermediate (Fig. 4c, occupancy ratios of conformer A and B were 70% and 30%, respectively). The conformer A was major and interacted with the reducing end of chitohexaose via a water molecule, but the protonation state was not clear. Therefore, both deprotonated and protonated Glu315 with conformer A were examined in the MD simulations. In one of the four MD simulations with deprotonated Glu315, the chitohexaose chain spontaneously moved in the forward direction (Fig. 4d, lower right, and Supplementary Movie 2). The bound chitohexaose moved

**Table 1 Statistics of data collection and refinement of the crystal structures of SmChiA with various bound oligosaccharide lengths**

|  | SmChiA-Slide-C6 (PDB ID: 5Z7M) | SmChiA-Slide-C5 (PDB ID: 5Z7N) | SmChiA-Slide-C4 (PDB ID: 5Z7O) | SmChiA-Slide-C3 (PDB ID: 5Z7P) |
|---|---|---|---|---|
| **Data collection** | | | | |
| Space group | $C222_1$ | $C222_1$ | $C222_1$ | $C222_1$ |
| Cell dimensions | | | | |
| $a, b, c$ (Å) | 131.41, 200.73, 59.54 | 131.11, 200.36, 59.51 | 131.05, 200.10, 59.41 | 131.86, 199.80, 59.60 |
| $\alpha, \beta, \gamma$ (°) | 90.00, 90.00, 90.00 | 90.00, 90.00, 90.00 | 90.00, 90.00, 90.00 | 90.00, 90.00, 90.00 |
| Resolution (Å) | 44.1-2.0 (2.12-2.0) | 44.1-1.7 (1.81-1.7) | 44.0-2.0 (2.12-2.0) | 44.2-2.0 (2.12-2.0) |
| $R_{sym}$ or $R_{merge}$ | 9.3 (53.7) | 9.8 (73.1) | 7.2 (40.5) | 8.7 (44.8) |
| $I / \sigma I$ | 16.5 (3.7) | 12.7 (2.5) | 17.6 (4.4) | 17.5 (4.5) |
| Completeness (%) | 99.2 (98.3) | 98.8 (96.8) | 99.9 (99.3) | 99.9 (99.7) |
| Redundancy | 7.7 (7.6) | 7.7 (7.6) | 7.6 (7.5) | 7.6 (7.5) |
| **Refinement** | | | | |
| Resolution (Å) | 2.0 | 1.7 | 2.0 | 2.0 |
| No. reflections | 135,242 | 250,646 | 146,556 | 149,277 |
| $R_{work}$ / $R_{free}$ | 15.2/18.4 | 17.4/19.7 | 16.6/19.7 | 15.5/18.8 |
| No. atoms | | | | |
| Protein | 4267 | 4267 | 4267 | 4267 |
| Ligand/ion | 278 | 248 | 251 | 77 |
| Water | 741 | 676 | 551 | 862 |
| $B$-factors (Å$^2$) | | | | |
| Protein | 30.3 | 26.3 | 36.0 | 29.3 |
| Ligand/ion | 40.8 | 33.3 | 56.2 | 45.0 |
| Water | 45.1 | 39.1 | 48.4 | 45.4 |
| R.m.s. deviations | | | | |
| Bond lengths (Å) | 0.010 | 0.006 | 0.005 | 0.016 |
| Bond angles (°) | 1.176 | 1.018 | 0.758 | 1.349 |

forward almost half *N*-acetyl glucosamine ring after sliding movements (Supplementary Figure 11a, at 25 ns and 250 ns indicated by arrows), and the first and second *N*-acetyl glucosamine rings from the reducing end was stacked on Phe396 and Trp275 during forward sliding, respectively. In the other three MD simulations, the chitohexaose chain moved backward due to the failure of stacking between the reducing end ring and Phe396 (Supplementary Movie 3). These results suggested that Phe396 assists forward sliding, in addition to stabilisation of the Michaelis complex at subsite +2. The deprotonated Glu315 did not interact with the chitin chain (Supplementary Figure 11b) and did not inhibit/assist forward sliding. On the contrary, the chitohexaose chain slid backward in both of the two MD simulations with protonated Glu315 (Fig. 4d, lower left, Supplementary Figure 11c and Movie 4). The protonated Glu315 formed hydrogen bond with the oxygen atom of *N*-acetyl group, and the distance between protonated oxygen atom of Glu315 side chain and oxygen atom of acetyl group became almost constant at 2.7 Å after backward sliding (Supplementary Figure 11d at ~13 ns indicated by an arrow). Then, first *N*-acetyl glucosamine ring of the reducing end stably stacked on the Trp275. Coupled with the hydrogen bond formation, backward sliding occurred rapidly (Supplementary Figure 11c). These results show that the decrystallised chitin chain bound with SmChiA can slide forward and backward rapidly by the thermal fluctuation (Brownian motion). Therefore, activation energy of chain sliding should be small and we concluded that decrystallisation is the rate-limiting step of processive movement of SmChiA.

**Binding free energy estimation by MD simulation**. We termed the intermediate structure after backward sliding as Chain-twisted

state (Fig. 5a, left), wherein one chitobiose unit of the bound chitohexaose withdrew from the catalytic site as compared with the Michaelis complex (Fig. 5a, right). Next, we estimated binding free energy ($\Delta G$) of decrystallised chitin chain with SmChiA in the Chain-twisted and Michaelis complex states, to understand how SmChiA counterbalances free energy increase caused by decrystallisation of single chitin chain. For estimation, we also performed all-atom MD simulation of the Michaelis complex (Supplementary Figure 11e and Supplementary Movie 5). The binding free energy ($\Delta G$) was estimated as a summation of internal, van der Waals, electrostatic interaction energies, and polar and non-polar solvation energies using the molecular mechanics/generalised Born surface area (MM/GBSA) method (Supplementary Figure 11f)[32]. We used MM/GBSA method, because it has an advantage that the binding free energies of each subsite can be calculated separately. The subsites in the Chain-twisted state were shown as −1' and −2', because the positions of substrates are slightly different from those in the Michaelis complex. The lowest binding free energy of −11.7 kcal mol$^{-1}$ was obtained at subsite −1 of the Michaelis complex, consistent with the smallest root-mean-square fluctuation in the previous study, indicating the most stable binding subsite[33]. The binding free energy at the product-binding sites (subsite +1 and +2) of the Michaelis complex was −16.0 kcal mol$^{-1}$, similar to that (−11.1 kcal mol$^{-1}$) of the most studied cellulase from an ascomycete *Trichoderma reesei* (TrCel7A) calculated by the free-energy perturbation/Hamiltonian replica exchange MD method[34]. Large binding free energy at the product-binding site is also consistent with the previous biochemical study of SmChiA proposing the importance of strong binding at the product-binding site[30]. The summations of binding free energy at Chain-twisted state −1' and −2', and Michaelis complex +2 to −2 subsites were estimated as −9.3 and −32.4 kcal mol$^{-1}$, respectively, indicating the Michaelis complex as −23.1 kcal mol$^{-1}$ more stable (Fig. 5a). On the

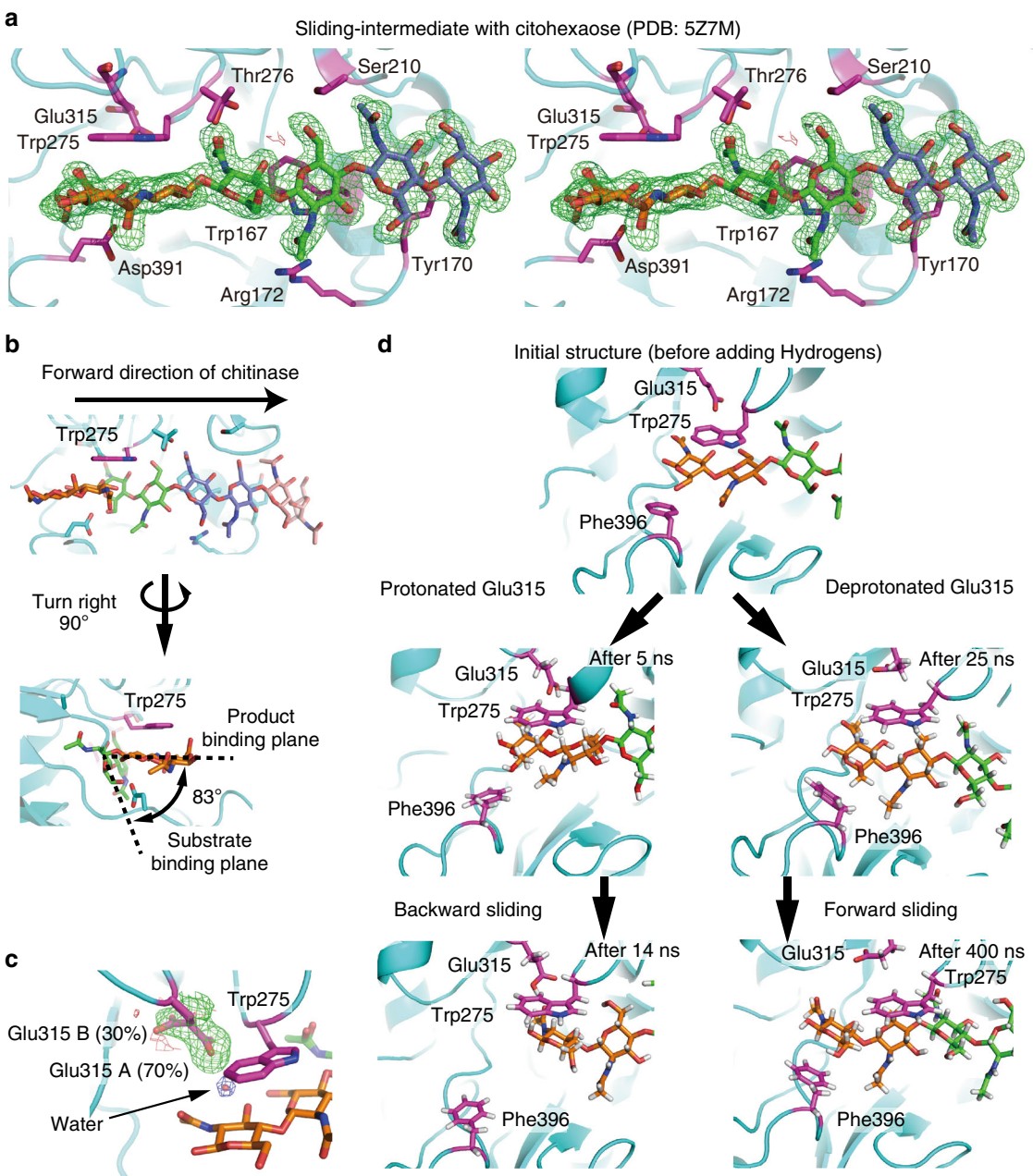

**Fig. 4** Structural and MD simulation analyses of chain sliding of SmChiA. **a** Cross-eyed stereo view of the omit map of chitohexaose molecule in the Sliding-intermediate structure (PDB ID: 5Z7M) shown at $3\sigma$; green: positive; red: negative. Substrate-interacting amino acid residues are shown by purple. **b** Product- and substrate-binding planes in the Michaelis complex-like structure of inactive D313A mutant (PDB ID: 1EIB); inter plane angle is 83°. **c** Omit map of Glu315 side chain in the Sliding-intermediate structure shown at $3\sigma$; green: positive; red: negative. $2F_{obs} - F_{calc}$ map of water molecule is also shown in $1.5\sigma$ (arrow). **d** Snap shots of MD simulation showing forward and backward chain slidings from the Sliding-intermediate structure. Structures before (25 ns) and after (400 ns) forward sliding with deprotonated Glu315 are shown on the right, and those before (5 ns) and after (14 ns) backward sliding with protonated Glu315 are shown on the left. Trp275, Glu315, and Phe396 were shown by stick. Structures were aligned with Trp275 as the centre of view

contrary, the equilibrium constant between the Chain-twisted (backward-stepped state; state *E* in Fig. 3f) and the Michaelis complex (forward-stepped state; state *A* in Fig. 3f) estimated from the rate constants of single-molecule imaging was 1.1 (58.5 s$^{-1}$/54.6 s$^{-1}$), indicating similar free-energy levels (difference of $-0.04$ kcal mol$^{-1}$ at 25 °C). This large discrepancy between the values estimated by MD simulation and by single-molecule imaging indicates that there is a working mechanism compensating the free-energy differences between the forward- and backward-stepped states as discussed below.

## Discussion

Our high-precision single-molecule imaging directly verified processive movement of SmChiA with 1 nm forward and backward steps (Fig. 2). This step size is consistent with length of reaction product chitobiose, indicating that SmChiA recognises each chitobiose unit and repeats the cycle of processive catalysis. Moreover, the time constant of substrate-assisted catalysis (2.9 ms, Fig. 3f) is shorter than that of product release (6.8 ms) and much shorter than that of decrystallisation (17.1 ms). It is known that SmChiA shows significantly higher

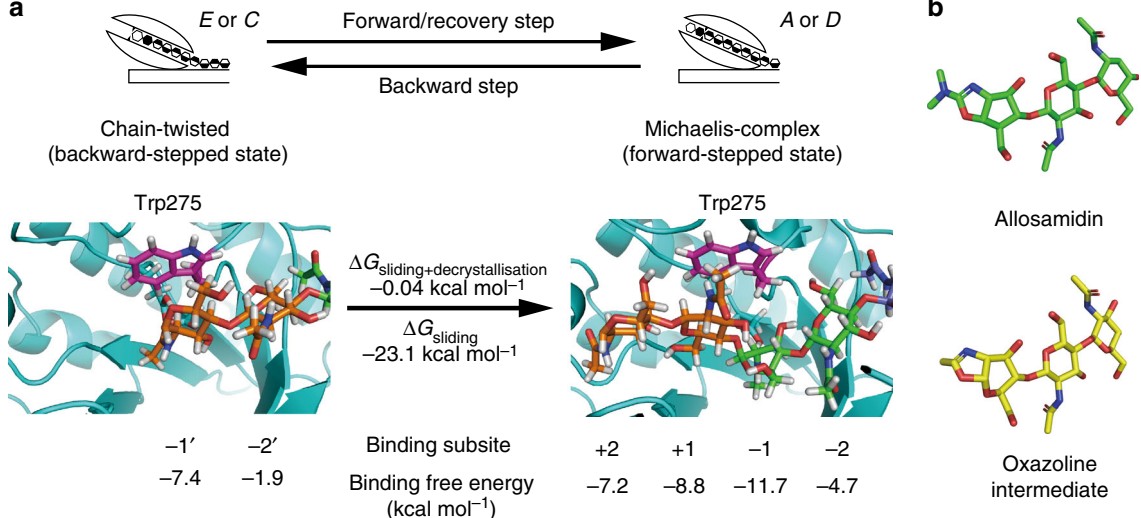

**Fig. 5** Binding free energy and structural similarity of intermediate to inhibitor. **a** Binding free energies at each subsite of the Chain-twisted (left, backward-stepped state) and the Michaelis complex (right, forward-stepped state). Corresponding chitobiose units in each state are shown in the same colours. **b** Crystal structure of allosamidin bound to SmChiA (top, PDB ID; 1FFQ) and expected structure of oxazoline intermediate (bottom). Green or yellow: carbon; red: oxygen; blue: nitrogen atoms

hydrolysis rate against 4-methylumbelliferyl chitobioside than that of analogues without C2 $N$-acetamide group[35]. Furthermore, higher hydrolysis rate of chitobiose than that of cellobiose catalysed by hydrochloric acid has been reported[36]. These results indicated that the $N$-acetyl group in oligosaccharide accelerates the glycosidic bond cleavage. Thus, well-designed substrate-assisted catalysis of SmChiA will also support fast degradation of chitin.

The free-energy differences between the Chain-twisted (backward-stepped) and the Michaelis complex (forward-stepped) states estimated by MD simulation ($-23.1$ kcal mol$^{-1}$) and single-molecule imaging ($-0.04$ kcal mol$^{-1}$) seems inconsistent apparently. However, because SmChiA moves forward and backward with chitin chain decrystallisation and recrystallisation, respectively, this inconsistency can be solved by assuming that single chitobiose unit decrystallisation requires ~23 kcal mol$^{-1}$ and the total free energies between forward- and backward-stepped states are almost similar. This value is larger than the free energy required for chitin decrystallisation in the absence of chitinase, estimated using the MD umbrella sampling ($-6$ to $-8$ kcal mol$^{-1}$ per chitobiose)[37], whereas similar to that estimated by the MM/Poisson–Boltzmann surface area (PBSA) method ($-8.7$ kcal mol$^{-1}$ per $N$-acetyl glucosamine, thus $-17.4$ kcal mol$^{-1}$ per chitobiose)[38]. In the former case, position-difference of the single chitin chain on crystal surface was discussed and, in the latter case, effect of the crystal size was taken into consideration. The absolute value of free energy required for decrystallisation during the chitinase movement is still an open question for further studies.

From the results of this study, we conclude that SmChiA operates as a Brownian ratchet that compensates chitobiose unit interaction from the crystal surface to its binding site. The Brownian ratchet as an operation mechanism is also supported by the lack of large conformational changes among the structures of SmChiA without substrate (PDB ID: 1CTN) and bound with different substrates (Supplementary Figure 12). Furthermore, the results of MD simulation suggest that switching of interaction between the sugar ring at the reducing end of the chitin chain and the aromatic amino acid residues (Trp275 and Phe396) supports sliding movement. As a Brownian ratchet motor, SmChiA movement is forward-biased through fast substrate-assisted

catalysis. After forward steps, SmChiA forms the Michaelis complex (state $A$ in Fig. 3f) and substrate-assisted catalysis (state $A$ to $B$) competes with backward step (state $A$ to $E$). Probability of the substrate-assisted catalysis is 86.3%, determined by the rate constant ratio of substrate-assisted catalysis (345 s$^{-1}$) and backward step (54.6 s$^{-1}$). This value is consistent with the forward step ratio (83.5% = 69.3% + 14.2%), independently estimated from the step size distribution (Fig. 2d). Following substrate-assisted catalysis and product release, SmChiA cannot move backward, because the chitobiose unit for backward movement was removed from the chitin chain. These results indicate burnt-bridge Brownian ratchet as modelling the biased movement (Supplementary Movie 6), which constitutes a universal mechanism for natural and artificial molecular motors decomposing the rail, such as collagenase and a DNAzyme-based walker[39–41].

It has been shown previously that the moving velocity of TrCel7A is ten times slower than that of SmChiA[4,5], although similar burnt-bridge mechanisms are expected to be used[42]. TrCel7A hydrolyses glycosidic bonds of cellulose by two glutamate residues. Especially, TrCel7A forms reaction intermediate with covalent bond between one glutamate and substrate, and this step is proposed as the rate-limiting by QM/MM simulation[43]. Although further studies are required for complete elucidation of the rate-limiting step of TrCel7A, slower catalysis of TrCel7A than that of SmChiA will correspond to less forward-biased movement of TrCel7A than of SmChiA. In fact, it has been reported that TrCel7A showed 1.9-fold larger backward step ratio (32%)[44] than that of SmChiA (16.5%, this study). Thus, slow moving velocity of TrCel7A would be partially due to slow catalysis and high backward step ratio. In contrast to TrCel7A, SmChiA does not form covalent bond with substrate during substrate-assisted catalysis[36]. Instead, the oxazoline (or oxazolinium ion) intermediate will exhibit high affinity towards SmChiA, because it has a similar structure to allosamidin, high affinity, and strong chitinase inhibitor (Fig. 5b)[45]. High-affinity reaction intermediate will facilitate bound state stability similar to covalent-bonded intermediate of TrCel7A. However, it is also known that the oxazoline intermediate is unstable and readily hydrolysed in

acidic condition[46–48]. Therefore, rapid hydrolysis of oxazoline intermediate will also occur in SmChiA without next-step inhibition, contributing to fast reaction processivity and movement. The operation mechanism of SmChiA revealed in this study will be helpful not only to engineer chitinases[30,49] and cellulases[50] for more efficient chitin and cellulose degradations, but also to design fast-moving artificial molecular motors[41,51].

## Methods

**Enzyme and substrate preparation**. The gene encoding SmChiA was synthesised based on the amino acid sequence (PDB ID: 1EIB) with reverse mutation (A313D) to the active wild-type and codons optimised for *Escherichia coli*. A six-histidine tag (His6-tag) was also added to the C terminus of SmChiA. For single-molecule observation, free cysteine residues (G157C and V351C) were introduced by swapping the region between AflII and AleI of wild-type SmChiA to the synthesised gene fragment with these mutations. A recognition site for factor Xa (IEGRFGG) was inserted between the C terminus of SmChiA and the His6-tag by PCR (Supplementary Table 1). The prepared gene was inserted into pET27b (Novagen) using NdeI and NotI. *E. coli* Tuner (DE3) (Novagen) was transformed with the plasmid by electroporation. Colonies, grown on Luria broth (LB)-kanamycin plates after 14 h incubation at 37 °C, were inoculated in 20 ml LB medium containing 50 μg ml⁻¹ kanamycin. After 30 min pre-incubation at 30 °C, 10 ml culture medium was inoculated in 1 litre LB medium with the same concentration of kanamycin and incubated at 30 °C with 130 r.p.m. shaking. When the $OD_{600nm}$ reached 2.0, the medium was cooled on ice for 10 min and 400 μl of 1 M IPTG (Wako) was added. The medium was further incubated at 20 °C for 16 h with continuous shaking at 130 r.p.m. Cells were collected by centrifugation at $3000 \times g$ for 10 min and 10 g of cells were suspended in 100 ml of 100 mM sodium phosphate buffer (pH 7.0) containing 100 mM sodium chloride. The enzyme was extracted by sonication and supernatant was collected after centrifugation at $8000 \times g$ and $30,000 \times g$. Crude enzyme was loaded onto a Ni-NTA agarose column (3 ml of column volume, Qiagen) and washed by 25 ml of 100 mM sodium phosphate buffer (pH 7.0) containing 100 mM sodium chloride with 0 and 50 mM imidazole. SmChiA was eluted by the same buffer with 100 mM imidazole and concentrated to 500 μl using ultrafiltration (Vivaspin 20 with 30 kDa-cut membrane: Sartorius). SmChiA was loaded onto Superdex200 10/300 GL equilibrated with 50 mM Tris-HCl buffer (pH 8.0) after 10 mM dithiothreitol (DTT) reduction for 1 h at 25 °C and a portion of the collected enzyme (100 μl, 99.3 μM) was mixed with 2 μl of 100 mM CaCl₂ and treated by 0.1 mg ml⁻¹ Factor Xa protease (NEB) at 23 °C for 16 h. Digested SmChiA was reduced with 10 mM DTT for 1 h at 25 °C and further purified with Suprdex200 10/300 GL equilibrated with 100 mM sodium phosphate buffer (pH 7.0) containing 100 mM sodium chloride. Collected enzyme was separated into two tubes and mixed separately with 3 mol amounts of Cy3-maleimide (GE Healthcare) or biotin-PEAC5-maleimide (Dojindo). After reaction at 25 °C for 1 h, unreacted reagents were removed by NAP-10 column (GE Healthcare) equilibrated with 50 mM sodium phosphate buffer (pH 6.0) and stored at −80 °C. SmChiA concentration was calculated from the absorbance at 280 nm; enzyme extinction coefficient was $\varepsilon_{280nm} = 107,050$ M⁻¹ cm⁻¹. The enzyme labelling ratio was calculated from the absorbance at 280 nm and 550 nm, extinction coefficient of enzyme described above, and Cy3-maleimide ($\varepsilon_{280nm} = 12,000$ M⁻¹ cm⁻¹ and $\varepsilon_{550nm} = 150,000$ M⁻¹ cm⁻¹).

To solve the substrate-bound structures of SmChiA, five point mutations (D313A, K369M, F396A, W539A, and E540M) were introduced. This mutant was designed to be catalytically inactive and form the intermediate complexes with substrates by reducing the affinity of the Michalis complex. First, the plasmid including a part of SmChiA and the fragment containing D313A, W539A, and E540M mutations were amplified (Supplementary Table 1). The fragments were connected by In-fusion cloning kit (Clontech). Prepared plasmid was used as the template, and K369M and F396A mutations were introduced by PCR. The linear DNA fragment was circularised by In-fusion cloning kit. The enzyme was purified by Ni-NTA agarose column in the same manner and buffer was exchanged to 20 mM Tris-HCl (pH 8.0). Collected enzyme was loaded into a column equilibrated by 20 mM Tris-HCl (pH 8.0) and eluted by a linear gradient of sodium chloride from 0 to 500 mM. Collected enzyme was further purified by Suprdex200 10/300 GL equilibrated with 10 mM sodium phosphate buffer (pH 7.0) containing 100 mM sodium chloride. Enzyme was stored at 4 °C until crystallisation.

The SmChiA mutant for chitotetraose hydrolysis (W167A) was produced and purified by the same methods (Supplementary Table 1). The W167A mutation (at subsite −3) was introduced to suppress the non-productive binding of chitotetraose.

Highly crystalline β-chitin was purified from tubes of *Lamellibrachia satsuma* according to the method described below[5]. Never dried tubes were cut into ~3 cm sections by scissors and incubated in 1 M NaOH for a night at 25 °C. The tubes were washed by ultrapure water and treated with 0.3% NaClO₂ in 100 mM sodium acetate (pH 4.9) for 3 h at 70 °C. Treated tubes were washed by ultrapure water and incubated in 0.1 M HCl at 80 °C for 20 min. The chitin was washed again by ultrapure water and homogenised by Physcotron blender (Microtech). The paste of

chitin was collected after centrifugation at $10,000 \times g$ for 10 min and kept at 4 °C. Thirty grams of never dried chitin paste and 120 ml of 6 N HCl and 30 ml of ultrapure water were mixed and incubated for 8 h at 80 °C with agitation by a blade at 300 r.p.m. The remaining crystalline chitin was washed by ultrapure water until the value of pH became 7. The suspension was kept at 4 °C before use.

**AuNP coating**. AuNP (400 μl, 40 nm in diameter) (BBI Solutions) was mixed with the same amount of 10 mM sodium borate buffer (pH 8.0) containing 0.5% (w/v) Tween20 and incubated at 25 °C with gentle rotation. PEG5000-SH (25 μl, 8 mg ml⁻¹) (Nanox) and biotin-PEG3400-SH (6.25 μl, 3 mg ml⁻¹) (Nanox) dissolved in ethanol were mixed with the suspension and incubated at 70 °C for 2 days. The reaction mixture was centrifuged at $10,000 \times g$ for 3 min and supernatant was removed. The pellet was suspended in 1 ml of 10 mM sodium borate buffer (pH 8.0) containing 0.5% (w/v) Tween20 and washed additional five times. Biotin-coated AuNP was re-suspended in 1 ml of the same buffer and mixed with 250 μl of 1 mg ml⁻¹ streptavidin (Prospec), dissolved in the same buffer. The mixture was incubated for 3 h at 25 °C, mixing gently. Streptavidin-coated AuNPs were washed in the same manner and suspended in 20 μl buffer and stored at 4 °C for a maximum of a week before use. Molar concentration of AuNP was estimated from absorbance at 520 nm ($\varepsilon_{520nm} = 8.411$ nM⁻¹ cm⁻¹).

**Single-molecule imaging**. TIRDFM[27] was used for observation with modification to improve the localisation precision. TIRDFM was constructed using an inverted microscope (IX71, Olympus) equipped with an ultra-stable microscope stage (Model KS-O, Ikeda-rika) and an objective lens (APON 60XOTIRF, ×60, numerical aperture (NA) = 1.49, Olympus) set on a vibration-isolated table (h-TDIS- 168LA(Y)OU2, Herz). A 532 nm optically pumped semiconductor laser (Sapphire 532-100 CW CDRH, Coherent) was introduced from the side port of the microscope and reflected towards the peripheral, high NA region of the objective lens using a perforated mirror with an elliptical anti-reflecting surface (minor axis: 6.0 mm, major axis: 8.5 mm, circular window viewed from the optical axis) to form the evanescent field at the interface of cover slip and sample solution. The light scattered by a single AuNP was collected through the perforated mirror and a five times extender lens (TS-EX-50, Sugitoh), and projected onto a high-speed CMOS camera (SA-5, Photron) customised to turn off the cooling fan during image acquisition. Image pixel size was 66.7 nm in both x- and y-axes, and the size of illumination field was 38 μm in diameter. Image acquisition was performed by commercial software (PFV, Photron) and analysed by ImageJ (NIH).

To measure localisation precision of AuNP attached to a glass surface, bare AuNP suspension was loaded into the flow cell made by two cover slips (24 × 32 mm and 18 × 18 nm, thickness 0.12–0.17 mm, Matunami glass) incubated in 10 M KOH for 16 h at room temperature and rinsed by ultrapure water until pH became neutral[23]. AuNPs were immobilised on glass and unbound particles were washed away using 200 μl of 100 mM sodium chloride. Scattering images of AuNP were recorded at 1.0 ms temporal resolution with various laser powers (1.0–8.0 μW μm⁻²), and with 7.0 μW μm⁻² laser power at various frame rate (1.0–0.2 ms temporal resolutions). Localisation precision of the x-axis (horizontal axis) and y-axis (vertical axis) was calculated as standard deviations of centre positions of five-particles estimated by two-dimensional (2D)-Gauss fitting of particle images for 2 s (Supplementary Figure 2a, b). Signal intensities of 40 nm AuNP, observed at 0.5 ms temporal resolution with 7.0 μW μm⁻² laser power, were analysed as average values of 2000 frames in 8 pixels × 8 pixels region of interest including single spot. Values were normalised by the maximum intensity of 12 bit camera (4095 counts) and distribution was fitted by the Gaussian function (Supplementary Figure 2c). Distribution of length of crystalline chitins was analysed using ImageJ with the scattering images at 2000 fps with 7.0 μW μm⁻² of 532 nm laser. Distribution was fitted by the log-normal distribution function (Supplementary Figure 3b).

For single-molecule imaging of SmChiA movement, streptavidin-coated AuNPs and SmChiA were mixed at final concentrations of 50 and 100 pM, respectively, in 20 μl of 10 mM sodium phosphate buffer (pH 6.0) for 1 h at 25 °C. The cover slip, washed by 10 M KOH, was spun at 3000 r.p.m. using a spin-coater (1H-DX II, Mikasa), and 60 μl of 0.01% crystalline chitin suspension was dropped at the centre. Then, a flow cell was prepared with the chitin-coated cover slip. The flow cell was filled with 20 μl of 10 mM sodium phosphate buffer (pH 6.0) and set on the microscope. The stage was equilibrated for 10–20 min to minimise the stage drift with a hydrated stage cover for preventing drying. After stopping the cooling fan of the CMOS camera, the AuNP-SmChiA mixed solution was loaded into the flow cell and images were recorded at 0.5 ms temporal resolution with 7.0 μW μm⁻² laser intensity at 25 °C. Images of moving and non-moving molecules in the same field of view were obtained simultaneously. The signal intensities of moving AuNPs were analysed by same way to AuNPs immobilised on the glass, except the values were analysed as the average of total moving frames. The moving distance was calculated from the difference of positions between initial frame and final frame. Distribution was fitted by the single exponential decay function (Supplementary Figure 3c). For the observation in D₂O, 1 M sodium phosphate buffer (pD 6.0) was prepared (shown as pH 5.6 in a pH metre using a glass electrode[52]). The SmChiA solution was diluted 10⁵-fold and AuNP was washed twice by 10 mM sodium phosphate (pD 6.0) and the flow cell was filled by the same buffer. Other observation conditions were the same as the H₂O condition. Images were recorded

at 0.5 ms temporal resolution. Trajectories were fitted by the linear function and the average velocities were calculated as the slope.

**Trajectory and step size analysis**. The centre of the AuNP image was estimated by 2D-Gaussian fitting to the image using the same algorithm as for the analysis of stepping movements of kinesin-1[28]. The axes, which are parallel and perpendicular to the long axis of crystalline chitin, were termed the on-axis and off-axis, respectively. Raw trajectory ($X_{raw}$, $Y_{raw}$) was rotated using the following equations: $Y_{on-axis} = \cos\theta \cdot X_{raw} + \sin\theta \cdot Y_{raw}$ and $X_{off-axis} = -\sin\theta \cdot X_{raw} + \cos\theta \cdot Y_{raw}$, in which $\theta$ is the angle between the on-axis and $X_{raw}$-axis. Noise of the rotated trajectory ($X_{off-axis}$, $Y_{on-axis}$) was reduced by using median filtering (current ± 2 ms). Steps in on-axis displacement were detected by the step-finding algorithm developed by Kerssemakers et al.[31]. The trajectories, which showed clear contrast of evaluation values ($S$-value) and sufficient precision to distinguish 1-nm steps during pauses (Supplementary Figure 5), were used for step size and subsequent dwell time analyses (see below). Distributions of sizes of forward and backward steps were fitted using the Gaussian functions, and the event ratio was calculated from the area of the fitted curves.

**Dwell time analysis**. For dwell time analysis, steps were categorised as (1) 1 nm forward and (2) backward steps, (3) recovery steps from the backward state, and (4) 2 nm forward steps. These steps were separated according to the step sizes and the state of the previous one step. For example, forward steps with the size of 0–2 nm just after forward steps are categorised as 1 nm forward steps, backward steps just after forward steps are categorised as backward steps, forward steps just after backward steps are categorised as recovery steps, and forward steps with size larger than 2 nm and just after forward steps are categorised as 2 nm forward steps. Continuous backward steps (backward steps just after backward steps) were rare ($N = 17$) and insufficient events were observed to determine a reliable time constant. Distribution of the dwell time of 1 nm forward steps was fitted by the convolution of two exponential decay functions; $y = a \cdot [\exp(-b \cdot x) - \exp(-c \cdot x)]$, where $a$, $b$, and $c$ are constants. Distributions of the dwell time of other steps were fitted by the single exponential decay function: $y = a \cdot \exp(-b \cdot x)$, where $a$ and $b$ are constants. As described below (Supplementary Figure 6 and 7), the first bins of the histogram were excluded from fitting.

**Generation of simulated trajectory of SmChiA movement**. Step trajectories were simulated using an algorithm based on the kinetic Monte Carlo method[53]. The algorithm consists of randomly choosing the next reaction and dwell time of the reaction. Reactions considered here are forward (1.1 nm or 2.2 nm), backward, and recovery steps. The recovery step always follows the backward step. The input parameters used in the simulation are rate constants (shown in Fig. 3f), step sizes, ratio of these steps, and localisation precision observed in the experiment. For choosing the next reaction (step), we randomly chose to reproduce the ratio of the forward and backward steps (note that this procedure differs from the original method). For choosing the dwell time, we randomly chose $\Delta t = \ln(1/r)/k$, where $r \in (0,1]$ is a uniform random number and $k$ is a rate constant of the step, to reproduce the dwell time distribution. Then, Gaussian noises with means set to zero were added to the step trajectories. At every pause, SD of the Gaussian noises was randomly chosen from experimentally observed distribution (Supplementary Figure 4c). Simulated and experimental step size distributions were nearly identical (Fig. 2d and Supplementary Figure 5e). Furthermore, 2 nm step peaks were only found in simulated trajectories with 2.2 nm steps, indicating that double-sized steps were not observed through insufficient temporal resolution (0.5 ms). Dwell time distributions of simulated steps before noise addition were compared with those of algorithm-detected steps from simulated trajectories (Supplementary Figure 6a and b). Very short dwell time (first histogram bin) event numbers were over- and under-counted through over- and under-fittings, respectively (Supplementary Figure 6c and d). Nevertheless, simulated time constants were well reproduced by the distribution curve fitting without the first bin. Thus, the dwell time distributions for experimental results were also fitted without the first bin.

**Kinetic isotope effect to soluble oligosaccharide hydrolysis**. Chitotetraose (Megazyme) was dissolved in milliQ water or $D_2O$. SmChiA (W167A mutant, final 10 nM) was incubated with 5–540 μM of chitotetraose at 25 °C for 2 min in 20 mM ammonium acetate buffer pH 6.0 or pD 6.0. Reaction was stopped by mixing acetonitrile to 70% (v/v) and 50 μl of mixture was loaded onto high-performance liquid chromatography (JASCO) equipped with NH2P-4E column (Shodex). Produced chitobiose and remained substrate were separated by 70% (v/v) acetonitrile mixed with milliQ water and detected by absorbance at the wavelength of 210 nm. Products were quantified by the standard curve. Plots of turnover against chitotetraose concentration were fitted by the Michaelis–Menten equation with substrate inhibition: $y = k_{cat} \cdot [S] / (K_m + [S] + [S]^2 / K_{si})$, where $k_{cat}$, $K_m$, and $K_{si}$ are the turnover, the Michaelis constant, and the substrate inhibition constant, respectively, and $[S]$ is chitotetraose concentration. KIE was calculated as the ratio of $k_{cat}$ in $H_2O$ to that in $D_2O$.

**X-ray crystallography**. The 1 μl drop of 15 mg ml$^{-1}$ SmChiA mutant (D313A, K369M, F396A, W539A, and E540M) was mixed with the same volume of 700 mM

sodium citrate (pH 6.4) containing 10% (v/v) MeOH and 5% (v/v) glycerol in a 96-well sitting-drop plate (Greiner Bio-one). The drop was equilibrated with 100 μl of 700 mM sodium citrate (pH 6.4) containing 10% (v/v) MeOH and 5% (v/v) glycerol at 20 °C for a week. The powders of chitohexaose, chitopentaose, chitotetraose, and chitotriose were dissolved in 1 μl of 700 mM sodium citrate (pH 6.4) containing 30% (v/v) glycerol. The biggest pieces of fan-shaped crystal were dipped in these solutions for 30 min at 20 °C and flush cooled by cryo-stream (100 K). Diffracted spots of 1.12 Å X-ray by the crystals were measured for 360° with 1° oscillation at BL2S1 in Aichi Synchrotron Radiation Center[54]. Data reduction was done by X-ray detector software and phase was determined by molecular replacement with Phaser[55] using SmChiA (PDB ID: 1EIB) as a template. Structure models were refined and improved by Phenix refine[56] and Coot[57]. No Ramachandran outliers were observed in the four structures.

**All-atom MD simulation**. Initial structure was prepared from the Sliding-intermediate structure (Fig. 4a, PDB ID: 5Z7M). All-atom MD simulations were performed using Amber 2016 software[58]. The force fields of ff14SB and GLY-CAM_06j were used for protein and sugar, respectively, and the TIP3P model was used for water. Basically, the protonation states of amino acids were analysed using the PDB2PQR server[59], except that both deprotonated and protonated states were used for Glu315. Each system of chitohexaose-complexed protein was solvated with water, and counter ions were added to preserve electric neutrality, where chitohexaose occupied subsites −1' to −6' (virtual) in the Sliding-intermediate structure. Then, for each system, 300-step energy minimisation, 500-ps equilibration under NVT condition (300 K), and 500 ps equilibration under NPT condition (300 K, 1 bar) were conducted with the heavy-atom positional restraints of 10 kcal mol$^{-1}$ Å$^{-1}$. Four and two production runs were conducted for structures with deprotonated and protonated Glu315, respectively. The initial complex with deprotonated Glu315 showed a spontaneous forward sliding, and that with protonated Glu315 showed a spontaneous backward sliding as described in the main text.

**Estimation of binding free energy**. To compare substrate binding in the Chain-twisted (backward-stepped) state with that in the Michaelis complex (forward-stepped) state, starting structure of the Michaelis complex state was constructed from an inactive E315Q mutant structure (PDB ID: 1EHN)[60], where the aspartic acid (Asp313) and the N-acetyl side chain of the sugar at subsite −1 were reoriented manually to mimic the Michaelis complex and the sugars at subsites −5 and −6 were removed to adjust lengths of bound oligosaccharide between the Chain-twisted and the Michaelis complex states, and the mutation in the structure was reverted to the wild-type (Q315E). Then, similar procedures described above (solvation, minimisation, equilibration) were conducted. Finally, the production runs were conducted twice (50 ns each for the Michaelis complex). Using the 100 ns trajectories of the Chain-twisted and the Michaelis complex in total (where the last 50 ns periods were used in the Chain-twisted state), the binding free energy ($\Delta G$) at each subsite was estimated by the MM/GBSA method[32,61] in which the MM energies (including the internal, van der Waals, electrostatic interactions), polar solvation, and non-polar solvation energies were calculated. Because Glu315 in the Michaelis complex has to be protonated for catalysis, the Chain-twisted state with protonated Glu315 was used for estimation.

## Data availability
Data supporting the findings of this manuscript are available from the corresponding authors upon reasonable request. The coordinates of SmChiA with bound oligosaccharide are deposited in the Protein Data Bank with the PDB ID 5Z7M, 5Z7N, 5Z7O, 5Z7P.

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

## Acknowledgements

We thank Y. Niitani (Nikon) and M. Tomishige (Aoyama Gakuin Univ.) for providing the step-finding algorithm plugin for ImageJ, H. Ueno (Univ. Tokyo) for discussion about preparation of streptavidin-coated gold nanoparticles, Y. Okuni and M. Yamamoto for helping sample preparation, and K. Tokuyasu (National Food Research Institute) for helpful discussion. X-ray diffractions were measured at BL2S1 in AichiSR. This study was supported by Grant-in-Aid for Scientific Research on Innovative Areas "Molecular Engines" (grant number JP18H05424 to R.I.), Grants-in-Aid for Scientific Research (grant numbers JP17K19213, JP16H00789, JP16H00858, and JP15H04366 to R.I., and

JP15H06898, JP17K18429, and JP17H05899 to A.N.) from the Ministry of Education, Culture, Sports, Science, and Technology (MEXT), Japan, Advanced Technology Institute Research Grants 2015 (RG2709 to A.N.), Interdisciplinary Research Promotion Project (J281002 to K.O.), and cross-disciplinary study (01311805 to A.N.) of the National Institutes of Natural Sciences (NINS). K.O. is partly supported by Building of Consortia for the Development of Human Resources in Science and Technology, MEXT, Japan.

## Author contributions

R.I. and A.N. conceived, designed, and supervised the project. A.N. prepared the samples and carried out single-molecule, biochemical, and structural analyses. K.O. simulated trajectory of movement and carried out MD simulation for chain sliding. T.F. and M.S. carried out MD simulation for chain sliding and binding free-energy estimation. All authors discussed the data and wrote the manuscript.

## Additional information

**Competing interests:** The authors declare no competing interests.

 

