## [Peer Review File · Nature Communications]

Reviewers' comments:

Reviewer #1 (Remarks to the Author):

The paper from Iino and co-workers examines the chitinase A from *Serratia marcescens*, denoted here as SmChiA, using nanoparticle-labeling strategy along with microscopic tracking analyses to attempt to elucidate the rate-limiting steps in chitin catalysis. The experimental work is interesting and seems to be rigorously executed, but the interpretation is perhaps a bit oversold. The paper could also benefit substantially from an English language editing service.

One question I have regards the deuterated water experiment. Does water take part in both steps of the SmChiA mechanism? For the glycosidic bond cleavage step, I do not think that it does. If that is the case, then how confident are the authors that the deuterated water experiment described on page 3 is capturing what the authors are proposing in the paper?

The computational work in the manuscript is rather cursory. The method to compute the binding free energy (is it ΔE or ΔG ? This is also completely unclear) is quite simplistic. Much better would be to do more rigorous calculations such as those pioneered by Roux or others. Computational work to decrystallize chitin has been reported by Beckham et al., so the number of 23 kcal/mol that the authors assume can be checked against previous work.

The authors state that because ChiA binds ligands in different positions based on crystal structures, then there must be some lack of preference of binding. This is not supported by much more than structural data, which do not necessarily reflect binding free energies. This should be modified. It is also completely unsupported by the (albeit cursory) binding free energy calculations discussed on page 5. Overall, the experimental results are interesting, but I think that the use of such simple calculations combined with rather unfounded assumptions calls into question the validity of the interpretation.

There are many other groups who have attempted to probe similar questions, and I find the citation list very lacking; indeed, the authors very much need to interpret their data in light of a significant body of previous work on both chitinases and cellulases. The groups of Igarashi, Westh, and Valjamae have done considerable amounts of work in TrCel7A's catalytic cycle, and these papers are barely mentioned at all. Work from the Eijsink group is also incredibly important in the context of how SmChiA works, its structure, its processive action, and the catalysis of chitin, but most of these seminal papers are not cited. Computational work from the NREL group (Beckham et al.) is also not discussed, including the decrystallization free energy of chitin and binding free energy of polysaccharides ligands to enzymes.

On a minor note, the authors remark that their study will help in the design of non-natural cellulases and fast-moving artificial molecular motors. Have non-natural cellulases or chitinases ever been engineered before or have cellulases informed how to design molecular motors? The authors are by no means the first to study how cellulases and chitinases work, so I feel like this comment is more sales than science.

Reviewer #2 (Remarks to the Author):

Comprehension of molecular mechanisms of chitinase and cellulase enzymatic action is crucial not only for enzymatic depolymerisation of the recalcitrant biopolymers, but also for better understanding of their roles as molecular motors. The manuscript presents exciting results about a mechanism by which chitinases transform Brownian motions in a unidirectional translational dislocation of the enzymes and measure mechanical kinetic parameters of these events on a single molecular level. Structural studies and molecular dynamics simulations further assist in comprehension of interactions and 3D constraints that are important for this process. The

manuscript presents important findings, is consistent and clearly written and I have only minor suggestions with respect to the work and its presentation:

1. Page 6, lines 19-20: The authors report that they introduced two free cysteine residues (G157C and V351C) for single molecule observations. I suppose that this has been done to facilitate labelling. However, why they introduced five mutations (D313A, K369M, F396A, W539A, E540M; page 11, lines 25 and 26) is less clear. Could the authors explain why this has been done and how molecular, structural and enzymatic properties of the mutant were different from those of the wild type enzyme.
2. Fig4, page 22. "Difference map of chitohexaose molecule in Sliding-intermediate (bottom) is also shown at 3 sigma; green: positive; red: negative." What kind of difference map is this? An omit map? I can't really see the active site interactions very well here. Can the structure of the complex be shown in stereo (may be in Supplementary Materials?).
3. Page 32, Extended data Figure 9. Panel A: The same question as before: Could the authors show the panes A in a stereo mode and explain what kind of maps are presented?

Reviewer #3 (Remarks to the Author):

Title: A Brownian monorail operated by fast catalysis after peeling rail from polymer crystal

Journal: Nature Communication

Manuscript No: NCOMMS-18-12695-T

Review:

The manuscript reports the chitinase Brownian monorail movement with high precision at single molecule level. The chemical reaction behind the forward and backward chitinase movement was analyzed along with binding free energy. The catalytic reaction and recrystallization of the chitinase was analyzed with dwell-time. Finally, the SmChiA-chitinase was compared with TrCe17A cellulose velocity to confirm the trajectory and step analysis movement. I would recommend its publication in Nature Communication after major revision, as the strategy demonstrates the precise monitoring of fast moving biomolecules.

Below are several comments to improve the manuscript:

1. The AuNP movement on glass surface was reported to be 0.3 nm at 0.5 ms temporal resolution. However, the movement will vary for single nanoparticle to cluster of nanoparticles. Therefore, the data should be more precise. The author should discuss whether the observed movement is for single nanoparticle, if not, then how far the velocity varies with respect to the cluster of nanoparticle.
2. Moving velocities of AuNP-SmChiA was reported to be slower than cy3-SmChiA. Does the velocity differ due to the weight of nanoparticle? In this case, the authors need to discuss whether the nanoparticle weight has any influence on the velocity of the movement. Will there be any difference in velocity, if a more small size nanoparticle used as the label instead of 40 nm gold nanoparticle.
3. How author concluded the unidirection movement of gold nanoparticle shown on Movie S1 was along the chitin fibrils? What is the total length of Chitin fibrils? How long the movement of the gold nanoparticle on chitin fibrils was monitored?
4. The scattering wavelength of 40 nm gold nanoparticle should not interfered by the background scattering of the chitinase A. Therefore, more optical properties should be discussed in detail about the label and substrate.
5. Comparative experimental evidences should be added for the slower velocity of cellulose in TrCe17A and SmChiA.

Response to the Reviewers' comments:

First of all, we sincerely appreciate the valuable comments raised by the Reviewers. According to the comments, we have extensively revised manuscript. All changes are highlighted with red color in the revised manuscript. With the helpful comments by the Reviewers, we believe that the manuscript has been significantly improved and is suitable for publication in Nature Communications.

The Reviewers' comments are shown in *Italic*.

Reviewer #1 (Remarks to the Author):

The paper from Iino and co-workers examines the chitinase A from Serratia marcescens, denoted here as SmChiA, using nanoparticle-labeling strategy along with microscopic tracking analyses to attempt to elucidate the rate-limiting steps in chitin catalysis. The experimental work is interesting and seems to be rigorously executed, but the interpretation is perhaps a bit oversold. The paper could also benefit substantially from an English language editing service.

Response: We appreciate critical comment by the Reviewer #1. According to the comment, we have revised the manuscript extensively. We have also used an English language editing service. We have attached the certificate of proofreading as the supplementary information for reviewer only.

One question I have regards the deuterated water experiment. Does water take part in both steps of the SmChiA mechanism? For the glycosidic bond cleavage step, I do not think that it does. If that is the case, then how confident are the authors that the deuterated water experiment described on page 3 is capturing what the authors are proposing in the paper?

Response: We also do not think that water is involved in the glycosidic bond cleavage step, as described in the previous and revised manuscripts (Fig. 1 and Supplementary Figure 1). On the other hand, we think that water takes part in the hydrolysis step of

oxazoline or oxazolinium ion intermediate. Therefore, in the manuscript, we described these two steps collectively as the “substrate-assisted catalysis.” In our study, we obtained two time constants (2.9 ms and 23.9 ms) before forward 1-nm step in H₂O, and found that only shorter time constant lengthened 3.5-fold (10.1 ms vs 2.9 ms) whereas the longer time constant did not change (24.8 ms vs 23.9 ms) in D₂O (Fig. 3a and b). From the result of the kinetic isotope effect (KIE), we concluded that the shorter time constant (2.9 ms) observed in H₂O corresponds to the substrate-assisted catalysis which consists of glycosidic bond cleavage and hydrolysis of oxazoline intermediate. Furthermore, in the revised manuscript, we also investigated the KIE for hydrolysis of water-soluble chitotetraose by SmChiA (Supplementary Figure 8e). Please note that in this experiment, we used a mutant of SmChiA (F167A, mutation at subsite -3), because wild-type SmChiA showed strong substrate inhibition. As result, the KIE (2.3-fold) comparable to the crystalline chitin (3.5-fold) was observed. This result supports the notion that the KIE observed with the crystalline chitin corresponds to the substrate-assisted catalysis, but not to decrystallisation or chain sliding.

The computational work in the manuscript is rather cursory. The method to compute the binding free energy (is it delta E or delta G? This is also completely unclear) is quite simplistic. Much better would be to do more rigorous calculations such as those pioneered by Roux or others. Computational work to decrystallize chitin has been reported by Beckham et al., so the number of 23 kcal/mol that the authors assume can be checked against previous work.

Response: We have clarified that we calculated delta G (ΔG) according to the molecular mechanics/generalised Born surface area (MM/GBSA) method described previously (Gohlke H. et al. *J. Mol. Biol.* 2003 (Ref 32); Kollman P. A. et al. *Acc. Chem. Res.* 2000 (Ref 61)) (page 8 line 12, page 8 line 17). We described more detailed methods and references for the MM/GBSA methods (page 20, line 7). As pointed out by the Reviewer #1, the MM/GBSA method used in our study is less rigorous than the free energy perturbation (FEP) and thermodynamic integration (TI) methods used by Roux and others. However, the MM/GBSA method shows similar precision to the umbrella sampling (US) method which has been used by Beckham et al. for the decrystallisation

analysis of chitin (Beckham G. T. *et al. J. Phys. Chem. B*, 2011 (Ref 37)). Furthermore, among these methods, only the MM/GBSA method has a merit to perform the energy decomposition analysis per residue/subsite. Therefore, we used the MM/GBSA method to analyze the binding free energy at each subsite. In the revised manuscript, this merit has been clarified (page 8 line 20). With this merit, we found large binding energy at product sites (subsite +2 and +1) (Fig. 5a), as suggested by previous biochemical assay (Kurasin M. *et al. J. Biol. Chem.*, 2015 (Ref 30)). Furthermore, according the comment by the Reviewer #1, we have described comparison of our results with previous studies (page 8 line 26, page 10 line 3).

The authors state that because ChiA binds ligands in different positions based on crystal structures, then there must be some lack of preference of binding. This is not supported by much more than structural data, which do not necessarily reflect binding free energies. This should be modified. It is also completely unsupported by the (albeit cursory) binding free energy calculations discussed on page 5. Overall, the experimental results are interesting, but I think that the use of such simple calculations combined with rather unfounded assumptions calls into question the validity of the interpretation.

Response: To further verify fast sliding of decrystallised chitin chain, we have added new results of MD simulations using the Sliding-intermediate with deprotonated Glu315 as the initial structure (Fig. 4 and Supplementary Figure 11). In one of the four MD simulations, the chitin chain slid forward spontaneously at the 25 and 250 ns simulation (Supplementary Figure 11a), and the Phe396 stuck on the reducing end of chitohexaose to assist forward sliding (Fig 4d right). In the other three MD simulations, the chitohexaose chain moved backward due to the failure of stacking between the reducing end ring and Phe396 (Supplementary Movie 3). The spontaneous backward sliding was also observed in MD simulations using the Sliding-intermediate with protonated Glu315 as described in the previous manuscript (Supplementary Figure 11c), because protonated Glu315 formed hydrogen bond with oxygen atom of N-acetyl group at the reducing end of chitin chain (Supplementary Figure 11d). These results indicate that sliding events are fast and not the rate-limiting of the movement. With these MD

simulations, we also showed the effect of hydrogen bond between Glu315 and N-acetyl group of chitin chain for sliding. We believe that these findings are important to understand the moving mechanism of SmChiA, and have described in the revised manuscript (page 7 line 8).

There are many other groups who have attempted to probe similar questions, and I find the citation list very lacking; indeed, the authors very much need to interpret their data in light of a significant body of previous work on both chitinases and cellulases. The groups of Igarashi, Westh, and Valjamae have done considerable amounts of work in TrCel7A's catalytic cycle, and these papers are barely mentioned at all. Work from the Eijsink group is also incredibly important in the context of how SmChiA works, its structure, its processive action, and the catalysis of chitin, but most of these seminal papers are not cited. Computational work from the NREL group (Beckham et al.) is also not discussed, including the decrystallization free energy of chitin and binding free energy of polysaccharides ligands to enzymes.

Response: We appreciate the comment. We have added the paragraph describing the processive action of cellulase and chitinase in the Introduction section, with the references from the research groups pointed out by the Reviewer #1 (Ref 13 to Ref 21, page 2 line15). Please note that we did not include papers focusing on the initial binding to or dissociation from the crystalline polysaccharide, because the main topic of our paper is the mechanism of processive movement of SmChiA as a molecular motor, but not the whole cycle of reaction including the initial binding and dissociation. To clarify the role of Trp275 of SmChiA, we also cited previous study by Valjamae group, showing the lower processivity of W275A mutant than that of wild-type (Kurasin M. *et al. J. Biol. Chem.* 2015 (Ref 30), page 7 line 6). We also described the series of studies from the NREL group (Beckham, Knott, Payne et al.) to discuss the binding and decrystallisaion energies of chitin (Jana S. *et al. J. Phys. Chem. B.* 2016 (Ref 33), Payne C. M. et al. *J. Am. Chem. Soc.* 2013 (Ref 34), Beckham G. T. *et al. J. Phys. Chem. B* 2011 (Ref 37), page 8 line 23, and page 10 line 3).

On a minor note, the authors remark that their study will help in the design of non-natural cellulases and fast-moving artificial molecular motors. Have non-natural cellulases or chitinases ever been engineered before or have cellulases informed how to design molecular motors? The authors are by no means the first to study how cellulases and chitinases work, so I feel like this comment is more sales than science.

Response: We appreciate the comment. According to the comment, we have revised the Abstract and Discussion sections. We have cited previous studies on engineered cellulases and chitinases (Kurasin M. *et al. J. Biol. Chem.* 2015 (Ref 30), Liu T. *et al. J. Biol. Chem.* 2017 (Ref 49), Taylor L. E. *et al. Nat Commun* 2018 (Ref 50), Page 11 line 20).

Reviewer #2 (Remarks to the Author):

Comprehension of molecular mechanisms of chitinase and cellulase enzymatic action is crucial not only for enzymatic depolymerisation of the recalcitrant biopolymers, but also for better understanding of their roles as molecular motors. The manuscript presents exciting results about a mechanism by which chitinases transform Brownian motions in a unidirectional translational dislocation of the enzymes and measure mechanical kinetic parameters of these events on a single molecular level. Structural studies and molecular dynamics simulations further assist in comprehension of interactions and 3D constrains that are important for this process. The manuscript presents important findings, is consistent and clearly written and I have only minor suggestions with respect to the work and its presentation:

Response: We sincerely appreciate high evaluation by the Reviewer #2.

1. Page 6, lines 19-20: The authors report that they introduced two free cysteine residues (G157C and V351C) for single molecule observations. I suppose that this has been done to facilitate labelling. However, why they introduced five mutations (D313A, K369M, F396A, W539A, E540M; page 11, lines 25 and 26) is less clear. Could the

authors explain why this has been done and how molecular, structural and enzymatic properties of the mutant were different from those of the wild type enzyme.

Response: We appreciate the important comment. The mutant (D313A, K369M, F396A, W539A, E540M) for X-ray crystallography was designed to observe less stable intermediate complexes with substrates than the Michaelis-complex. For this purpose, five amino residues in subsite +2 to -2 were mutated to alanine or methionine to reduce the affinity of the Michaelis-complex. These amino acid residues for mutation do not interact with chito-oligosaccharides in the substrate-enzyme complex (Supplementary Figure 10a). Furthermore, superposition of chito-oligosaccharides in the substrate-enzyme complex and wild-type SmChiA showed no steric hindrance (Supplementary Figure 10b). Therefore, we concluded that solved crystal structures are snap shots of intermediate structures during the processive movement. These explanations were added in the revised manuscript (page 6 line 20, and page 13 line 5).

2. Fig4, page 22. “Difference map of chitohexaose molecule in Sliding-intermediate (bottom) is also shown at 3 sigma; green: positive; red: negative.” What kind of difference map is this? An omit map? I can’t really see the active site interactions very well here. Can the structure of the complex be shown in stereo (may be in Supplementary Materials?).

Response: As the Reviewer #2 pointed out, the omit map of chitohexaose is shown in the previous Fig. 4a. We have revised the Fig. 4a to the stereo view for better understandings of readers. We have also revised the legend for Fig. 4a as follows: “Cross-eyed stereo view of the omit map of chitohexaose molecule...” (page 28 line 5).

3. Page 32, Extended data Figure 9. Panel A: The same question as before: Could he authors show the panes A in a stereo mode and explain what kind of maps are presented?

Response: We have revised the Supplementary Figure 9 to the stereo view and the legend as follows: “Cross-eyed stereo views of the omit maps of the crystal structures...”.

Reviewer #3 (Remarks to the Author):

Title: A Brownian monorail operated by fast catalysis after peeling rail from polymer crystal Journal: Nature Communication Manuscript No: NCOMMS-18-12695-T

Review: The manuscript reports the chitinase Brownian monorail movement with high precision at single molecule level. The chemical reaction behind the forward and backward chitinase movement was analyzed along with binding free energy. The catalytic reaction and recrystallization of the chitinase was analyzed with dwell-time. Finally, the SmChiA-chitinase was compared with TrCe17A cellulose velocity to confirm the trajectory and step analysis movement. I would recommend its publication in Nature Communication after major revision, as the strategy demonstrates the precise monitoring of fast moving biomolecules.

Response: We sincerely appreciate positive evaluation by the Reviewer #3.

Below are several comments to improve the manuscript:

1. The AuNP movement on glass surface was reported to be 0.3 nm at 0.5 ms temporal resolution. However, the movement will vary for single nanoparticle to cluster of nanoparticles. Therefore, the data should be more precise. The author should discuss whether the observed movement is for single nanoparticle, if not, then how far the velocity varies with respect to the cluster of nanoparticle.

Response: According to the comment of the Reviewer #3, we measured and compared signal intensities of bare 40-nm gold nanoparticle (AuNP) immobilised on glass and those of AuNP-labelled SmChiA moving on crystalline chitin. Distribution of signal intensities for bare AuNP was well fitted by the single Gaussian (Supplementary Figure 2c). This result indicates that observed AuNPs are single particles, because the signal

intensity will increase non-linearly with the sixth-power of the diameter if they form cluster (Dijk M. A. et al. PCCP 2006). Furthermore, signal intensities of AuNP-labelled SmChiA (0.36 ± 0.10 A.U., $N = 17$) was similar to those of bare AuNP immobilised on the glass (0.29 ± 0.19 A.U., $N = 267$, Supplementary Figure 2c). Therefore, we concluded that the moving SmChiA molecules were labelled by single AuNP particles. We have described these results and conclusion in the revised manuscript (page 4 line 5).

2. Moving velocities of AuNP-SmChiA was reported to be slower than cy3-SmChiA. Does the velocity differ due to the weight of nanoparticle? In this case, the authors need to discuss whether the nanoparticle weight has any influence on the velocity of the movement. Will there be any difference in velocity, if a more small size nanoparticle used as the label instead of 40 nm gold nanoparticle.

Response: In our previous and revised manuscripts, we described that there are no significant differences between moving velocities of AuNP-labelled SmChiA and Cy3-labelled SmChiA. As shown in Supplementary Figure 3b, the moving velocities of AuNP-labelled SmChiA and Cy3-labelled SmChiA were 51.9 ± 21.7 nm/s ($N = 58$) and 55.3 ± 18.4 nm/s ($N = 57$), respectively. We evaluated significant difference using F-test and t-test. F-value of the two sets was 1.38 and smaller than the F Critical One-tail (1.56) with the significance level of 0.05. Therefore, we checked the difference of the data using the t-test with assumption of equal variances. The t-value of the data was 0.89 and smaller than the T Critical Two-tail (1.98) with significance level of 0.05. These results indicate that moving velocities of AuNP-labelled SmChiA and Cy3-labelled SmChiA were not significantly different. The result of t-test was described in the legend of Supplementary Figure 3b. Please note that mass of single 40-nm AuNP is sufficiently small and the inertial force is negligible. Furthermore, viscous drag of water applied to single 40-nm AuNP is also sufficiently small and the viscous force is negligible.

3. *How author concluded the unidirection movement of gold nanoparticle shown on Movie S1 was along the chitin fibrils? What is the total length of Chitin fibrils? How long the movement of the gold nanoparticle on chitin fibrils was monitored?*

Response: We appreciate the comment. We used the 532-nm laser light for scattering imaging of 40-nm AuNP. Crystalline chitin can be also directly observed with our imaging system, although signal intensity is much lower than that of 40-nm AuNP (an example is shown in Supplementary Figure 3a). Therefore, we can easily distinguish if the AuNP binds on chitin and moves along chitin microfibrils. This point has been clarified in the revised manuscript (page 4 line 9). Furthermore, according to the comment by the Reviewer #3, we investigated distributions of run length of AuNP-labelled SmChiA and length of crystalline chitin (Supplementary Figures 3c and d, respectively). The average run length and average length of crystalline chitin were 86.2 nm and 3.5 μm , respectively. The value of the run length was comparable to that of Cy3-labelled SmChiA. We have described these results in the revised manuscript (page 4 line 19).

4. *The scattering wavelength of 40 nm gold nanoparticle should not interfered by the background scattering of the chitinase A. Therefore, more optical properties should be discussed in detail about the label and substrate.*

Response: At the wavelength of 532 nm we used for imaging, the scattering intensity of chitinase A is far weaker than that of 40-nm AuNP. Furthermore, the scattering intensity of crystalline chitin is also much weaker than that of 40-nm AuNP (Supplementary Figure 3a). By using this property, we can easily recognize binding and movement of AuNP-labelled SmChiA on crystalline chitin as described in the response to the comment #3 of the Reviewer #3 above and in the revised manuscript (page 4 line 9).

5. *Comparative experimental evidences should be added for the slower velocity of cellulose in TrCel17A and SmChiA.*

Response: The slower velocity of TrCel17A than that of SmChiA has been shown by the

previous single-molecule imaging with high-speed atomic force microscopy (Ref 4 and Ref 5), as described in the Introduction section (page 2 line 26). In the current study, we focused on the moving mechanism of SmChiA. We referred to the TrCel7A as an example of enzyme with similar moving mechanism, and proposed one possible mechanism of slower velocity than SmChiA. The detailed single-molecule analysis of the moving mechanism of TrCel7A will be a topic of future study. To clarify this point, we revised a sentence in the Discussion as follows: “Although further studies are required for complete elucidation of the rate-limiting step of TrCel7A, slower catalysis of TrCel7A than that of SmChiA will correspond to less forward-biased movement of TrCel7A than of SmChiA.” (page 11 line 7).

Other points of revision

1. We have changed the format of the manuscript according to the submission guidelines of Nature Communications.
2. We have added Supplementary Table 2 describing the primers used for generation of SmChiA mutants according to the submission guidelines of Nature Communications.
3. We have corrected the SD values of the moving velocities for Cy3-labelled SmChiA in H₂O and AuNP-labelled SmChiA in D₂O (Page 4 line 18, Supplementary Figure 3).
4. British English has been used in the revised manuscript according to the proofreading by the English language editing service.

REVIEWERS' COMMENTS:

Reviewer #1 (Remarks to the Author):

The authors have fully addressed my comments and concerns. I recommend that this manuscript now be published in Nature Communications.